# Mapping integrated implementation of Adapted Resource and Implementation Application (ARIA) and REDCap version hospital-based pediatric cancer registry (HBCR) in Ethiopia: An implementation Study

Diriba Fufa Hordofa[1]*, Yohannes Kebede[2], Mamude Dinkye[3], Tadele Hailu[3], Zewdie Birhanu[2], Megan C. Roberts[4], Victor Santana[5], Nickhill Bhakta[5]

1 Department of pediatrics and child health, Jimma University, Jimma, Ethiopia, 2 Department of Health, Behavior, and Society, Jimma University, Jimma, Ethiopia, 3 Department of pediatrics and child health, St. Paul Public Health Millennium Medical College, Addis Ababa, Ethiopia, 4 Division of Pharmaceutical Outcomes and Policy, Eshelman School of Pharmacy, University of North Carolina, Chapel Hill, North Carolina, United States of America, 5 Department of Global Medicine, St. Jude Children's Research Hospital, Memphis. Tennessee, United States of America

* diriba.fufa@ju.edu.et

## Abstract

This study presents the perceived implementability of the digital Hospital-Based Cancer Registry (HBCR) and the Adapted-Resource Implementation Application (ARIA) to enhance data systems and treatment standards at Pediatric Oncology unit (POU). A 2-year (2023–2025) implementation study on the integrated application of ARIA and HBCRs is being conducted at Jimma University Medical Center (JUMC) and St. Paul Hospital Millennium Medical College (SPHMMC). This article reports the formative assessment results, guided by the Consolidated Framework for Implementation Research (CFIR), involved eight focus group discussions, four in-depth interviews, and two co-design workshops with diverse healthcare providers and hospital management/leadership personnel. The integrated implementation of HBCR-ARIA was viewed as innovative and adaptable. Digital HBCR was perceived as more effective than manual methods for managing pediatric oncology data. Similarly, ARIA was perceived as effective and feasible for providing patient-specific standardized care. Workflows and responsibilities were co-defined separately for the respective POUs. The co-designed implementation strategy includes residents filling demographic and diagnostic information of patients' on HBCR interim document and then cross-checked by the pediatric Hematology and Oncology (PHO) fellows. The Medical Monitor (PHO senior) approves the validity of the document before entry into REDCap by the data clerk. ARIA is filled by PHO fellows and approved by second PHO fellow or PHO seniors based on the availability. Facilitators in both the inner (hospital) and outer (external) settings outweighed the barriers. Facilities and motivated human

**Data availability statement:** All relevant data are within the paper and its Supporting Information files.

**Funding:** This study was funded by St. Jude Children's Research Hospital and the American Lebanese Syrian Associated Charities (ALSAC). The Grant ID number for the funding is SJ 02-005-GPM-23. This funding was received by first author, DH. The funders had no role in study design, data collection and analysis, decision to publish, or preparation of the manuscript.

**Competing interests:** The authors have declared that no competing interests exist.

resources are in place to implement the digital HBCR and ARIA strategies at the respective POU. However, challenges such as inconsistent electric power, unreliable internet services, and logistic-supply issues. The implementation strategies for digitized HBCR and ARIA, co-designed to fit the specific contexts of two POU, appear promising but require further evaluation.

## Introduction

Ethiopia, the second most populous country in Africa with a population exceeding 115 million, faces significant challenges in pediatric oncology. Although exact data on the incidence of childhood cancer is lacking, estimates suggest over 10,000 new cases annually [1]. Despite the high incidence, only five hospitals across major cities in Ethiopia currently provide pediatric oncology care. These hospitals lack essential infrastructure such as hospital-based cancer registries (HBCRs) and standardized resource-adapted treatment guidelines, which are crucial for effective management and treatment of childhood cancer. Furthermore, the absence of a digitized data system and inconsistent performance indicators and registration variables exacerbates the problem [2,3]. This lack of standardized and reliable data hinders the ability to monitor treatment outcomes, evaluate program effectiveness, and make informed decisions for improving pediatric oncology care.

HBCRs are vital components of a comprehensive childhood cancer program. Patient Oncology summary sheet for demographics (POSSh-D) provide essential data on patient demographics disease characteristics and diagnostic work up done whereas pediatric oncology summary sheet for treatment and follow-up (POSSh-TF) - registers the treatment protocols, follow-up, and outcomes. HBCR is crucial for monitoring program quality and guiding future interventions [4,5]. However, in Ethiopia, the absence of HBCRs hinders the ability to collect and analyze this critical data, impeding efforts to improve treatment standards and patient outcomes.

Resource-stratified treatment guidelines are another key element in developing effective pediatric oncology programs in resource-limited settings [4,6,7]. Directly adopting treatment protocols from high-income countries can result in inappropriate and potentially harmful outcomes due to differences in healthcare resources and local conditions. Instead, adapted treatment guidelines are necessary to address the specific challenges faced by resource-constrained settings. Although organizations like the International Society of Pediatric Oncology (SIOP) and the International Network for Childhood Cancer Treatment and Research (INCTR) have developed adapted treatment regimens (ATRs), these guidelines often lack the comprehensive clinical decision-making support needed in low-resource contexts [8,9,10,11]. They typically do not provide detailed recommendations for diagnostic work-ups, dose modifications due to toxicities, and alternative regimens during drug shortages, which are common issues in low- and middle-income countries.

To address these gaps, St. Jude Children's Research Hospital, in collaboration with SIOP and other partners, developed the Adapted-Resource and Implementation

Application (ARIA) [12] The ARIA Guide is a web- and mobile-based clinical decision aid tool designed to support clinicians in low-resource settings by incorporating missing components from existing ATRs. This tool aims to enhance clinical decision-making with features such as treatment protocols adapted to local conditions, recommendations for handling drug shortages, and guidance on dose adjustments.

This implementation research (IR) [13,14] aimed to pilot the integration of ARIA and a HBCR REDCap version at two selected pediatric oncology units in Ethiopia. The study has pre-implementation, implementation and post-implementation phases and tasks [15]. The study utilized implementation science methods to evaluate how these digital solutions are integrated into resource-limited settings. The pre-implementation phase of the integration is being assessed using the Consolidated Framework for Implementation Research (CFIR) to identify barriers and facilitators [16,17].

With the goal to report comprehensive pre-implementation assessment findings, in this article, we aimed to answer the following key questions regarding the integrated implementability of REDCap version HBCRs, and ARIA. 1) To what extent is this digital solution perceived as appropriate and relevant in addressing the existing registration and treatment challenges of pediatric oncology units (POU)? 2) How is the ARIA and REDCap HBCRs implemented in the given context of the POU? 3) What implementation strategies are be used to implement integrating ARIA and HBCR in resource-limiting settings? 4) What are the barriers and facilitators within both the inner (hospital) and outer (external) settings of the POU, including implementers readiness, opportunities, and barriers?

## Methods

### Study setting and period

This IR was conducted in two POU in Ethiopia: St. Paul Hospital Millennium Medical College (SPHMMC) and Jimma University Medical Centre (JUMC) from August 1, 2023 to October 20, 2023. JUMC is a the only referral hospital treating pediatric oncology in south west Ethiopia with the catchment population of more than 20million, located about 355km south west of the capital, Addis Ababa. JUMC is affiliated with Jimma University and serving as a teaching hospital for medical students and residents. JUMC pediatric oncology unit established in 2016 with collaboration of Jimma University and The Aslan Project, and has only one PHO senior. JUMC also has PHO fellowship training program. Although it is one of the key centers for pediatric oncology (PO) in Ethiopia, it faces challenges such as lack of hospital-based cancer registries, limited standardized resource-adapted treatment guidelines, and insufficient digitized data systems. Despite these challenges, the center plays a crucial role in delivering pediatric cancer care (PCC) and is involved in efforts to improve childhood cancer treatment standards and infrastructure [18]. SPHMMC is one of a prominent healthcare institution located in Addis Ababa, Ethiopia. It is known for its comprehensive medical services and its role as a teaching and referral hospital. It also faces challenges similar to other centers like lack of HBCR and well established standardized resource adapted treatment guideline and shortage of PHO seniors, only two in the center [19]. Both JUMC and SPHMMC have PHO fellowship program. The pediatric oncology data was being collected using the excel sheet at both centers before the implementation of the REDCap version HBCR.

### Study design

Through its three phases, this IR focused on exploring perceived relevance of the digital solutions including needs and strategies (phase I: pre-implementation); implementing the strategies and plans (phase II), and evaluating the reach, acceptability, and sustainable adoption (phase III: post-implementation). The pre-implementation assessment and co-design activities were guided by CFIR (this article report this portion of the IR). At phase I, digital solutions and implementability of ARIA and HBCR integration thoroughly discussed at respective POU to identify needs, barriers and facilitators followed by implementation strategies co-design workshops.

## Population and data sources

The main data sources for this IR on integrated implementation of ARIA and HBCR were pediatric Hematologist/Oncologists, pediatric oncology fellows, pediatric residents, POU head nurses, data clerks, Monitoring and evaluation and quality officers, and Health Information Management System heads, ICT officers, as well as hospital leadership: chief executive director/Provost, medical director, chief Administrative and development director and head of pediatric oncology units from respective hospitals. A total of 28 respondents participated in the study.

## Data collection tool

Study tool (refer to S1 Text and S2 Text) separately developed for ARIA and HBCRs according to key constructs of CFIR framework; namely: (1) the intervention characteristics (e.g., relative advantage, adaptability, complexity and cost of the interventions); (2) outer setting(e.g., peer pressure, and external policies and incentives); (3) inner setting(e.g., structural characteristics, implementation climate and relative priority); (4) characteristics of individuals(e.g., knowledge and belief about the intervention, self-efficacy); and (5) implementation process (e.g., planning roles and responsibilities, standard operation procedures, work flow, key barriers and facilitators).

## Data collection

Four trained and experienced qualitative data experts collected the data. We conducted focus group discussions (FGD, n = 8) and key informant interviews (n = 4) with key potential implementers, supportive staffs, and hospital leadership at respective centers.

## Data analysis

The qualitative data were thematically analyzed using the CFIR framework with ATLAS.ti 7.5.18. Three experienced qualitative researchers with expertise in implementation science analyzed the data. While the coders used CFIR as a foundational guide, they applied open coding to identify emergent themes and sub-themes pertaining to HBCRs and ARIA. Implementation science experts reviewed and validated the coding process as well as the emergent themes to ensure methodological rigor. This analysis focused on the perceived innovativeness of digital solutions, implementation plans, and the barriers and facilitators related to implementation strategies, individual implementers, and the internal and external environments of POU for integrating ARIA and REDCap version HBCR. The sub-themes and key findings were supported by representative quotations. Code list emerged from the open coding annexed (S1 File).

## Ethical approval

The study was ethically approved by Jimma University's IRB. Oral informed consent obtained from the study participants after stating the purpose of the study, potential benefit and risk, duration of the interview, main topic of discussions, and the right to withdrawal from the study. The respondents were also granted the opportunity to ask any questions and summarize the key expectations from the interview before giving consent and proceeding to the study. The consent was witnessed by the respondents and stored in confidential cabinet with principal investigator.

The digital data for the REDCap-based HBCRs are securely stored on a local server managed by Jimma University. A dedicated ICT expert oversees data security, ensuring compliance with privacy protocols and safeguarding sensitive patient information throughout the implementation study. Data were confidentially and securely stored at Jimma University. Participant safety and rights protected during and post interviews. The study was conducted without any deviation from the approved protocol.

**Inclusivity in global research**

Additional information regarding the ethical, cultural, and scientific considerations specific to inclusivity in global research is included in the Supporting Information (S2 Checklist).

## Results

### Participant profiles

Table 1 has details of respondents' profile. Professionals from diverse discipline and positions participated in the study. Most participants had ≥ 2 years of experience working with POU or in the hospitals. These participants in their 24–50 years age had educational background ranging from diploma to sub-specialty and PhD holders.

### Perceived relevance of the intervention: integrated implementation of REDCap version HBCRs, and ARIA

The participants gave their experiences with the current registries and PO treatment methods, and their thoughts on REDCap version HBCR, and ARIA were communicated as relative advantages (relevance/effectiveness/benefits) and feasibility (difficulty/ease/practicality) as follows:

**Perceived relative advantage of the digital solution: effectiveness, efficiency and quality.** The discussants discussed that the existing electronic medical record in the hospital, and excel sheet that has been operational at POU had limited scope. Hence, the hospital and PO data were being mainly captured manually or in hard copy which was much difficult for data management. The possibilities of manual versioned outpatient and treatment follow-up data to be stored, accessed, traced and checked, monitored and validated, expediting service, interlinked across different units, and managed from a distance are limited—which this REDCap version HBCRs is believed to address. Furthermore, the ARIA was found helpful by Oncologists and hospital leaderships since it supports standardize and resource-adapted pediatric oncology care, so relevant in the deficiency of senior PHOs. ARIA simplifies the ordering of treatment protocols minimizing drug ordering error and saves time by automating dosage calculations that are context-specific and resource-adapted.

**Table 1. Participants profile, HBCRs-ARIA implementation study, 2025, Ethiopia.**

| Participant Role | Number | Education Level |
|---|---|---|
| Technical and support staffs | | |
| Pediatric Hematologist/Oncologists | 2 | Sub-specialty/PhD |
| Pediatric Oncology Fellows | 2 | Specialty (MD + Fellowship) |
| Pediatric Residents | 3 | MD (in training) |
| POU Head Nurses | 2 | BSc/Diploma in Nursing |
| General Pediatric Nurses | 2 | Diploma/BSc Nursing |
| Pharmacists | 2 | BPharm/MSc |
| Data Clerks | 2 | Diploma/Degree in Health IT/Records |
| M&E and Quality Officers | 3 | BSc/MSc in Public Health/M&E |
| HIMS Heads | 2 | BSc/MSc in Health Informatics |
| ICT Officers | 2 | BSc/MSc in IT/Computer Science |
| Hospital Leadership | | |
| • Chief Executive Director/Provost | 2 | PhD/MBA/MPH |
| • Medical Director | 2 | MD + Management Training |
| • Chief Admin/Development Director | 2 | MBA/MPA |
| • POU Unit Heads | 2 | MD/PhD (Pediatrics) |

"In our hospital; as a fellow physician, we register patients' data on hard copy format that has been created for this purpose, which is then collected by our data clerk to be copied into the computer. The REDCap version of the HBCRs would ease the registration and proper handling of the data" (FGD_HBCR, P4, JUMC)

"…Sometimes we found some prescription errors such as dose calculation. The integrated implementation of ARIA and HBCRs would improve standard of treatment and reduce such prescription errors as it automatically prescribes" (FGD_ARIA, P1, JUMC)

In addition to introducing REDCap version HBCRs, and ARIA, we also addressed other challenges related to PO data management. Since oncologic practice is still in its infancy across the pediatric oncology units (POU), key performance indicators (KPIs) for oncology were not standardized across POU, and also not included in DHIS-2, the hospitals' digital health information management system (HIMS) platform. As a result, these KPIs remained unreported to the broader health system and were only available for internal hospital use, if used at all.

**Perceived difficulty vs. easiness of implementation.** Most discussants shared the opinion that the implementation of the of ARIA and REDCap versions HBCRs in the respective POU was feasible. They had greatly made a strong emphasis on both adaptation by individuals and institutions to these beneficial ARIA and REDCap versions HBCR and through continuous follow-up and review meetings, more so in this digital era.

"This registration system uses to track the progress of patient, and treatment outcome, which is difficult to summarize from hardcopy registration. With the committed human resource we have, I hope that we can successfully implement this initiative and in the coming five years pediatric oncology unit has a sustained patient registry system which could be considered as exemplary in this hospital." (FGD_HBCR, P2, JUMC)

"I want to stress on the ownership issues. People has to see this initiative carefully it should be integrated and people should be convinced on its importance and why we do implement. So, to make this initiative a pattern in the routine activity, we have to repeatedly discuss with them. Continuous engagement, awareness and discussion should be made beyond an individual interest. Follow-ups and review meeting platforms should also be installed in the system. So, if you follow these recommendations, you will be successful and it is our common issue that we will work together." (FGD_HBCR, P4, SPHMMC)

However, assigning human resource for the REDCap versions and providing minimum capacity strengthening training is required to ensure its implementation.

"As a pharmacist, I would be happy if I take a role of crosschecking treatment protocol and dose before preparing the ordered medication; however, I do not think I will get enough time to take this additional responsibility as I am over-loaded with many tasks like preparing chemo medicines, receiving medicines from central store, and recording the medicines usage. If pharmacists should take additional responsibilities when the implementation of ARIA started, it is good to assign additional human resource." (FGD_ARIA, P2, JUMC)

### Implementation process and plan

This implementation process outlines the workflow or standard operating procedures (SOPs) and the implementers' roles and responsibilities in the integrated implementation of ARIA and REDCap version HBCRs.

*Standard operating procedures (SOPs): POSSh-D, ARIA, and POSSh-TF.* **Existing operation standard:** The extant workflow relies on manual registry data capture, which poses challenges in tracking, monitoring,

and summarizing key performance indicators—a limitation addressed by Step 5 of our intervention (the interim operation procedure described below). Furthermore, the absence of a standardized digital application for treatment guidance (Steps 6–9 in the proposed operation) introduces variability in standard care delivery. While our intervention is expected to enhance data quality and operational efficiency in specific aspects, we recognize that partial digitization—without full automation of registries or treatment follow-up—may initially require duplicate data entry (manual and REDCap and ARIA). This interim approach was necessitated by the pragmatic constraints of the trial, which focused on incremental improvements rather than full digitization.

### Interim standard operation procedure

Since the two POU are parts of comprehensive hospitals, some of the patients are admitted to the oncology unit after the suspicion of a cancer diagnosis at the emergency room (ER) or outpatient department (OPD). There are two interim documents created by the HBCR development team to facilitate easy data collection and ensuring hard coy back-up. These are named as pediatric oncology Summary sheet Demographic and Diagnostic (POSSh-D) and Pediatric oncology summary sheet for treatment and follow up (POSSh-TF). This step-by-step process outlines a thorough workflow for handling patients with suspected or confirmed cancer with rigorous digital data management and back-up documentation of HBCR information related to POSSh-D and POSSh-TF as shown Fig 1.

*Staffing and team makeup: roles and responsibilities*. Table 2 has details of staff roles and responsibilities. Key individuals for ARIA and REDCap version HBCRs include data clerks, medical monitors, PHO residents, nurses, pharmacists, supportive staff, hospital leadership, and project managers. Their roles are essential for the successful integration and management of these systems, covering data entry, patient care, drug administration, and project oversight.

"In my opinion, nurses will have many roles in the implementation process like, checking the availability of investigations, sending orders to laboratory, receiving result and informing physicians, measuring the weight, taking vital signs periodically, preparing patient for chemo administration and height of patients and so on. So far, there has been a huge gap regarding scaling the weight of patient and taking vital signs; sometimes, they order chemo for patient and tell us to discard the prepared chemo due to unstable vital sign of patient, I think they order medicine without checking the vital sign of their patient." (JUMC, FGD_ARIA, P1)

"To discuss the workflow, it starts from the point where patients first come to the hospital and registered at the card room. However, our real responsibility for this innovation started after patients admitted to our ward [oncology unit], at which residents and fellow physicians have the responsibility of filling and crosschecking data respectively. After data registered on paper and crosschecked by fellow physicians, collecting these paper based patient data and filling out on Red cap version of HBCRs is the responsibility of data clerk, finally, the ICT experts have a responsibility of overlooking or solving any ICT related issues." (JUMC, FGD_HBCR, P8)

**Integration and adaptation into existing workflow.** Integrating PO activities into existing hospital workflows required coordinated efforts in management structure development, protocol standardization, and alignment with health information systems. Success depends on effective training, continuous adaptation, and strong leadership engagement to ensure sustainability and scalability as follows.

*Structure and operations*: Collaboratively developed management structures, workflows, and Standard Operating Procedures (SOPs) tailored for the hospital setting. Designed tools for monitor and assessing the effectiveness of these ARIA and REDCap version HBCR.

## Pediatric Cancer Patient Workflow from Diagnosis to Completion

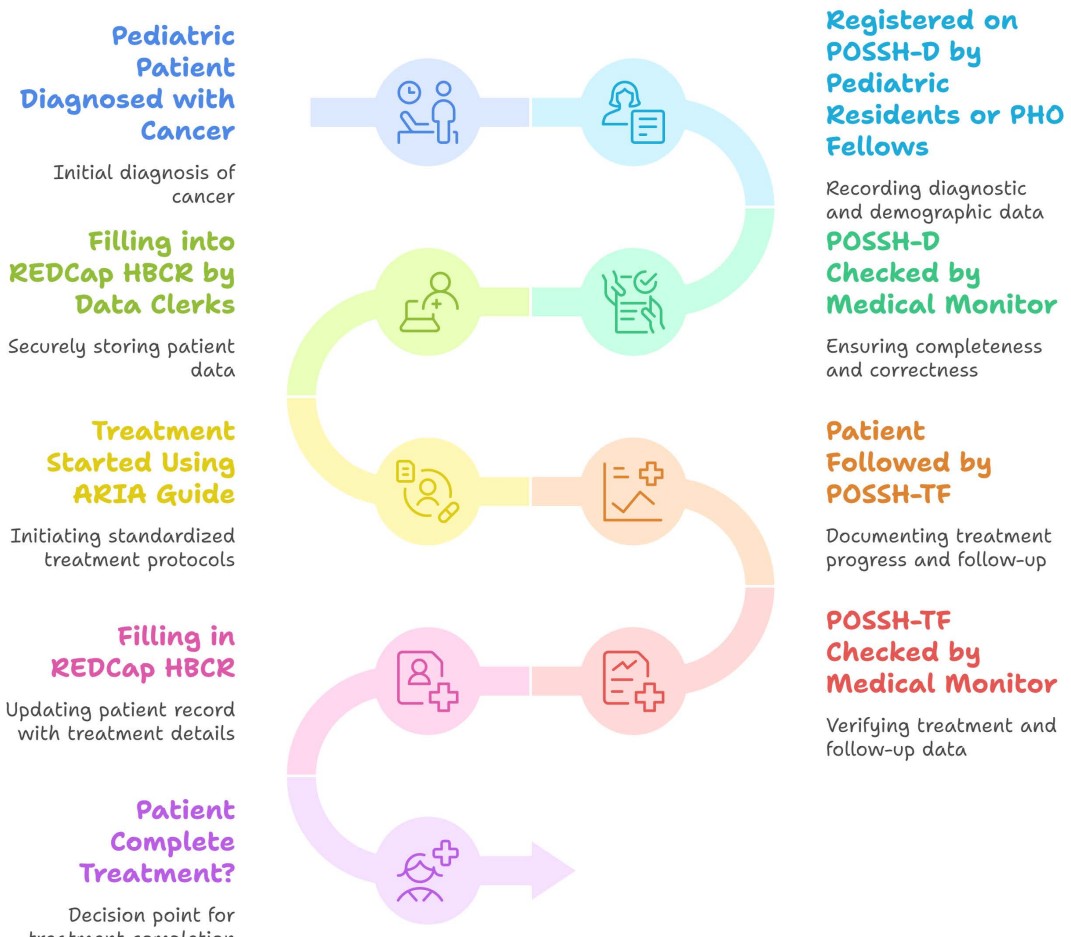

**Pediatric Patient Diagnosed with Cancer**

Initial diagnosis of cancer

**Registered on POSSH-D by Pediatric Residents or PHO Fellows**

Recording diagnostic and demographic data

**Filling into REDCap HBCR by Data Clerks**

Securely storing patient data

**POSSH-D Checked by Medical Monitor**

Ensuring completeness and correctness

**Treatment Started Using ARIA Guide**

Initiating standardized treatment protocols

**Patient Followed by POSSH-TF**

Documenting treatment progress and follow-up

**Filling in REDCap HBCR**

Updating patient record with treatment details

**POSSH-TF Checked by Medical Monitor**

Verifying treatment and follow-up data

**Patient Complete Treatment?**

Decision point for treatment completion

**Fig 1. Standard Operating Procedures for Integrated implementation of HBCRs-ARIA in Pediatric Oncology Units.**

*Integration with HIMS/DHIS-2 and adapting the intervention with the hospital's existing system*: Developed key performance indicators (KPIs) specifically for PO activities, ensuring seamless data exchange within hospitals and across POU. Cross linkage of the PO data captured through the REDCap version demographic, treatment and outcome registries across the HIMS of the respective POU through delineation of basic reportable KPIs and assignment of clear roles, proper engagement and awareness creation of the monitoring and evaluation and quality assurance teams. This would also ensure sustainable reporting and quality assurance of PO activities in general and REDCap versions in particular.

"Only PHO fellow physicians and data clerks have clear roles and responsibilities in the implementation of REDCap, in addition, nurse unit head has also some information about the initiative and plays roles like crosschecking the entered data by residents and fellow physicians. If we can integrate this initiative with DIHS2, we can also use roles and responsibilities defined in DIHS2 database. However, my question is that can we integrate red cap version of HBCRs? (FGD_ JUMC, P2).

**Table 2. Roles and Responsibilities of technical, supportive and administrative staffs, HBCRs-ARIA implementation study, 2024, Ethiopia.**

| Role | Key Responsibilities |
|---|---|
| Data Clerk | - Maintain data extraction forms (POSSh-D/TF)<br>- Input data into REDCap daily<br>- Verify form validity and ensure data security<br>- Conduct basic data analysis<br>- Report KPIs to M&E<br>- Attend review meetings and trainings |
| Medical Monitors | - Oversee HBCR/ARIA implementation<br>- Validate POSSh-D/TF forms<br>- Ensure accurate data entry in REDCap<br>- Confirm patient evaluations pre-chemotherapy<br>- Report challenges to project management |
| Residents/PHOs fellow and seniors | - Complete POSSh-D/TF with diagnostic/treatment data<br>- Use ARIA for chemotherapy prescriptions<br>- Verify weight, height, lab results for dosing |
| Nurses | - Record/update vital signs<br>- Add POSSh-D/TF to patient charts<br>- Transcribe ARIA drug orders<br>- Administer medications<br>- Report implementation challenges |
| Pharmacists | - Verify ARIA drug doses before dispensing<br>- Ensure safe drug mixing/delivery<br>- Coordinate with nurses/physicians<br>- Manage drug stock updates and Report ARIA issues |
| Technical Leadership (Department of pediatrics and child health and POU) | - Oversee HBCR/ARIA adaptation<br>- Ensure integration into workflows<br>- Collaborate on standardization & resource allocation<br>- Address implementation challenges |
| Hospital Quality Office | - Support quality assurance<br>- Monitor adherence to HBCR/ARIA<br>- Collaborate on sustainability strategies |
| Hospital M&E Unit | - Manage HBCR as routine data system<br>- Develop KPIs and review reports<br>- Ensure sustainable adaptation |
| Hospital Leadership (Administrative and supportive staffs) | - Ensure integration & sustainability<br>- Provide resources<br>- Promote HBCR/ARIA as routine practice<br>- Support capacity improvements |
| ICT Office | - Ensure internet/equipment availability<br>- Maintain server & technical support<br>- Assist with data summarization<br>- Ensure system interoperability |
| Project Management | - Oversee integration into hospital workflows<br>- Provide training & technical support<br>- Address implementation challenges<br>- Review progress & document experiences<br>- Submit quarterly reports |
| Mentors (Internal/External) | - Provide local & global guidance<br>- Advise on implementation research & best practices |
| HBCR Developers | - Experts from University of North Carolina and Ethiopia (server administrator, located in Tikur Anbessa Specialized Hospital) jointly lead the functionality and adaption REDCap and HBCR. Accordingly;<br>- Adapt REDCap based on user feedback<br>- Provide ongoing technical assistance |

*Training and co-design workshops*: Conducted comprehensive training sessions and co-design workshops for all relevant units, focusing on the use of REDCap version HBCR and ARIA systems. Discussed customized roles, address challenges, and align the units with the newly developed structures and operational protocols.

*Adaptation, sustainability, and scalability*: Engaged POU heads and hospital leadership to structurally integrate the new activities. A robust monitoring and evaluation system that ensure ongoing adaptation, long-term sustainability, and scalability of the integrated workflows and protocols established.

**Monitoring and evaluation process.** *Monitoring and evaluation tool and methods*: All PO patients' demographic, treatment, and outcome data will be managed through REDCap versions HBCRs and ARIA, according to SOPs, will be regularly recorded using M & E tool developed for this specific task at respective POU. In fact, only HL-Lymphoma will be treated through ARIA standard for the moment because the other disease is not finalized during the launch of the implementation study. These records are to be followed fortnightly during early months of the project and monthly, and quarterly. The Total number and percentages of specific pediatric cancer types admitted, number started treatment for curative and/or palliative intent, their outcomes such as abandonment, drop-outs, and deaths (defined per protocol) will be reported quarterly. Regular virtual discussion will be held among the core implementing team from respective hospitals/POU every fortnight in the first quarter and on monthly basis afterwards. The CIT will be made-up of data clerk, medical monitor, information and communication technology (ICT) experts, project PI and research assistant. Any technical and administrative challenges faced by CIT identified and addressed accordingly.

## Perceived barriers and facilitators of integrated implementation of ARIA and REDCap version HBCRs

Analyzing the perceived barriers and facilitators of integrating ARIA and REDCap version HBCRs involved examining the settings, implementation process, and implementers' contexts in which these systems are applied. Table 3 has details of barriers/challenges and enablers/opportunities in the inner and outer settings.

*Implementation challenges and opportunities*. The implementation of HBCRs and ARIA faces several challenges and opportunities. Interruptions in electricity and ICT infrastructure, while present, are gradually improving, offering a growing opportunity for enhanced implementation. However, compensation remains a critical challenge, as implementers require financial support that is not yet available. Ensuring the sustainability of REDCap and digitized registries, along with monitoring treatment with ARIA, is essential but faces obstacles due to the need for ongoing financial and operational commitment. Additionally, the poor documentation culture within the hospital further complicates implementation efforts, highlighting the need for improved practices and systems to support effective data management and usage.

**Key individual implementers: capacities, readiness, and expectations.** Reported barriers and enablers related to individual implementers' capacities, readiness, and expectations highlight several critical factors. Motivation has been reported as a significant enabler, with implementers showing strong readiness to engagement with the new systems. Training and capacity strengthening are also seen as positive factors, particularly due to the presence of a mentorship scheme that supports skill development. Conversely, reported barriers include inadequate human resources for sustainable data management and complications. Compensation issues are another challenge, as implementers expect financial support that is currently unavailable, placing a strain on sustainability and requiring the hospital to cover these expenses, particular for data managers. Additionally, the transition to digital solutions is hindered by a nascent culture of digitized registries and treatment, reflecting a barrier that needs further attention and development.

## Discussion

The following discussion points highlight key findings from this pre-implementation study and align them with relevant literature, providing a foundation for the implementation challenges and opportunities associated with ARIA and REDCap version HBCRs in Ethiopian POU.

**Table 3. Barriers and Facilitators, HBCR-ARIA implementation Study, 2024, Ethiopia.**

| Setting | Descriptions | Supporting Quotes |
|---|---|---|
| Inner: facilitators | POU, Hospital Structure, and Resources | |
| • Established POU with hospital priority | Dedicated space (OPD/training halls) and motivated staff (oncologists, nurses, pharmacists). | "This cancer registry initiative is a priority in our hospital... We have trained ICT staff to troubleshoot digital issues." (FGD, P2, SPHMMC) |
| • Digitization alignment | Hospital's intent to digitize records matches HBCR/ARIA goals. | "Software systems reduce manual work monotony and improve efficiency." (HMIS, JUMC) |
| • Strong M&E and quality control | Existing structures support integration and standardization. | "Digital HBCRs enhance traceability and treatment monitoring." (FGD, ARIA, JUMC) |
| • ICT expertise | JU and SPHMMC have technical capacity. | "We leverage local ICT skills for troubleshooting." (HMIS, SPHMMC) |
| Inner: barriers | | |
| • Staffing shortages | Lack of dedicated ICT staff, nurses, pharmacists; rotation of residents. | "Residents rotate frequently—training each batch is unsustainable." (FGD_HBCR, P1, JUMC) |
| • Ownership & compensation gaps | Staffs perceive HBCR/ARIA as unpaid extra work. | "People expect rewards for additional tasks... without incentives, it's a burden." (FGD_HBCR, P2) |
| • ICT infrastructure gaps | Compatibility issues, limited computers/tablets. | "Past digitization failed due to system limitations." (FGD, JUMC) |
| • AI risks in ARIA | Potential biases/errors in AI-driven dosing. | "ARIA's AI needs rigorous validation." (FGD, SPHMMC) |
| External: Opportunities | Health System, Socio-Cultural, and Political Conditions | |
| • National digital initiatives | Government supports healthcare digitization. | "Partners like [Aslan Project] may fund tablets/technical aid." (FGD_HBCR, P1, JUMC) |
| • Funder priorities | Expansion of ARIA to more cancer types. | "Funders focus on scaling ARIA's coverage." (FGD, JUMC) |
| External: Challenges | | |
| • Misaligned indicators | DIHS2-KIPs/ICD inconsistencies complicate integration. | "Pediatric cancer is lumped with adults—no dedicated MOH focus." (FGD, P1, SPHMMC) |
| • Medication shortages | Limited access to pediatric cancer drugs. | "Patient surges strain drug supplies." (FGD, JUMC) |
| • Political/financial instability | Budget cuts delay ICT/staffing sustainability. | "There is general budget deficits at country levels due to inflations" (JUMC) |

According to this study, participants perceived ARIA and REDCap version HBCRs as effective tools for improving data management and patient care. They highlight the limitations of existing manual systems and the benefits of automation in treatment protocols. Studies have also shown that digital systems and the integration of electronic health record systems, such as REDCap, has been associated with improved data accuracy, efficiency, and patient care as it enhanced data management capabilities and support clinical decision-making by standardizing treatment protocols and automating processes.. For instance, Koppel demonstrated how electronic systems can reduce data entry errors, while Lee found improved patient monitoring and coordination across oncology units. These benefits are particularly significant in resource-limited settings, where manual systems are prone to delays and fragmentation of care. In regional studies from sub-Saharan Africa, Khan also noted improvements in clinical efficiency and the quality of oncology records with structured electronic systems, suggesting ARIA and REDCap could produce similar advantages in Ethiopia [20,21,22].

Despite the perceived feasibility of implementing ARIA and REDCap version HBCRs, this study identified challenges including human resource limitations, ICT infrastructure issues, and integration difficulties with existing systems. Similar challenges have been documented in low and middle-income countries. For example, Yusof and Kluge describe the persistent struggle with limited technical infrastructure, unreliable power supply, and internet connectivity, all of which compromise the functionality of electronic health records. Furthermore, resistance to change among healthcare workers—often due to lack of familiarity or confidence with digital tools—has also been reported by Berg and Nguyen. As this study

observed, such barriers can undermine early adoption efforts and require targeted interventions including training, user support, and consistent leadership to overcome [23,24,25,26].

The current study found that successful integration of ARIA and REDCap version HBCRs requires the alignment of new systems with existing hospital workflows and health information systems. Leaders and staffs of relevant units across the hospitals engaged in the co-design workshop to devise workflow, share responsibilities, and established mechanisms for proper adaption of the intervention —such as developing regularly reportable KPIs with quality checks. Similarly, Liu and Anderson emphasized that digital health tools are most effective when adapted to fit within the operational realities of healthcare facilities. Greenhalgh underscored that co-design processes improve usability and increase user buy-in, leading to higher adoption rates. Integration with existing infrastructure—rather than replacing systems wholesale—was shown by Smith to improve both uptake and continuity, which aligns with this study's collaborative, phased approach [27,28,29,30,31,32,33].

This study revealed that regular monitoring and evaluation of workflow, activities, outputs, and outcomes through proper methods and indicators with properly trained personnel are necessary to track the effectiveness and sustainability of the implementation. IR team established core implementation team (CIT) from respective POU to discuss performance status and address challenges on fortnightly or monthly basis. Similarly evidences indicate effective monitoring and evaluation are essential for assessing the process/procedures and impact of new health interventions. Literature reinforces this need; for example, Peters, Bärnighausen and Kumar stressed the importance of ongoing review mechanisms to detect bottlenecks and mitigate risks early. Similarly, Higgins, Johnson, and Baker reported that iterative feedback loops and learning systems improve performance and ensure fidelity to program objectives. Monitoring tools not only guide adjustments but also provide transparency to stakeholders and support long-term investment [34,35,36,37,38,39].

The study identifies various internal and external factors affecting implementation, including resource availability, system compatibility, training and capacity strengthening, and external support. Research highlights the importance of contextual factors, such as resource availability and external support, in determining the success of health interventions. Damschroder and Greenhalgh emphasized that even well-designed interventions can falter without adequate alignment with the implementation environment. In a regional context, Rosen (2020) noted that leadership commitment, funding continuity, and integration with national health information systems are vital for sustaining digital innovation in oncology care. National frameworks and global guidance further validate the need to consider these determinants in scaling up [16,40,41,42,43,44].

Beyond this pilot phase, sustainability of the intervention may face several risks. Staff turnover could lead to loss of trained personnel, requiring ongoing capacity-building efforts to maintain competence and confidence in using the systems. System maintenance challenges—such as software updates, troubleshooting, and data security—are common hurdles that need designated technical support and budget allocations. Furthermore, transitioning from donor-supported phases to institutional or national funding is critical but often overlooked. These risks align with lessons from similar digital health initiatives in Ethiopia and other African settings [28,38]. To mitigate these, it is essential to embed digital systems within national eHealth architecture and health financing frameworks, and ensure local ownership. Broader implications for scaling nationally include harmonization with DHIS2 and other Ministry of Health platforms, inclusion in national cancer control strategies, and leveraging ARIA and REDCap data for surveillance, planning, and policymaking.

## Strength and limitations

This study demonstrates the feasibility of implementing an integrated digital pediatric cancer registry in resource-limited settings, emphasizing adaptability through stakeholder collaboration and scalable training approaches. However, its scope focused primarily on registry digitization and treatment scheduling, with less emphasis on improving resource availability (e.g., PO drug supplies) or advancing key pediatric oncology performance indicators critical to Ethiopia's service landscape. Time efficiency comparisons (e.g., manual vs. digital case entry) were not quantified, and duplicate data entry

risks persist during this transitional phase. Additionally, alignment with existing electronic medical records (EMRs) to avoid redundant entries was not fully explored. Despite these limitations, the interim system lays critical groundwork for full digitization, offering transferable lessons for similar low-resource contexts while highlighting the need for future expansions in resource tracking and system interoperability.

## Conclusions

Implementing the following evidence-informed recommendations and addressing the policy implications will enhance the integrated implementation and use of digital tools like ARIA and REDCap version HBCRs, leading to improved patient care and operational efficiency in POU. The integration of ARIA and REDCap version HBCRs is anticipated to improve data management and treatment efficiency in POU. While feasible, this integration faces challenges related to human resources, training, and system adaptation. Key barriers include limited electrical power, ICT infrastructure, and staffing issues, while facilitators include a supportive environment and motivated staff. Clear and adaptable SOPs and workflows were developed to guide the successful integration of these digital tools.

Comprehensive training opportunities should be implemented to equip staff with the skills to use ARIA and REDCap version HBCRs effectively, with ongoing support to tackle implementation challenges. Increased investment in ICT infrastructure, such as reliable internet and hardware, is necessary, alongside potential staff hiring or resource reallocation for digitized cancer registries and data system management. Standardized KPIs for pediatric oncology should be developed to ensure consistent data reporting, sharing, and integration across POU and broader health system. Robust monitoring and evaluation mechanisms are essential to assess the performance of the digital tools, with regular reviews to address any issues. It is also important to foster a sense of ownership among stakeholders in the inner (hospitals) and outer (external) settings and develop a sustainability plan that includes financial support and continuous capacity strengthening. Finally, efforts should be made to integrate ARIA and REDCap version HBCRs REDCap version with existing health management information systems (HMIS) for streamlined data management.

There should be advocacy for increased policy support and funding for digital health initiatives, particularly in cancer care. Policies that incentivize continuous training and capacity building for implementers are crucial. Additionally, promoting collaboration among hospitals, government agencies, and international organizations will support digital health innovations and resource sharing.

## Supporting information

**S1 Text. FGD and in-depth interview guide for HBCRs.**
(DOCX)

**S2 Text. FGDs and in-depth interview guide for ARIA.**
(DOCX)

**S1 File. Code list.**
(RTF)

**S1 Checklist. Inclusivity in global research checklist.**
(DOCX)

## Acknowledgments

We extend our sincere gratitude to all participants and interviewers for their invaluable contributions to this study. Special thanks to the PHO residents/fellows/seniors for their expertise and support. We also appreciate the leadership and support and POU staffs, and the Pediatric and Child Health Departments of JUMC and SPHMMC for their guidance,

resources, and crucial role in facilitating this research. We are also grateful to our mentors from Jimma University and external institutions for their insightful guidance and support throughout this research.

## Author contributions

**Conceptualization:** Diriba Fufa Hordofa, Zewdie Birhanu, Victor Santana.

**Data curation:** Diriba Fufa Hordofa.

**Formal analysis:** Yohannes Kebede.

**Funding acquisition:** Diriba Fufa Hordofa.

**Investigation:** Diriba Fufa Hordofa, Yohannes Kebede, Mamude Dinkye, Tadele Hailu.

**Methodology:** Diriba Fufa Hordofa, Mamude Dinkye, Tadele Hailu, Zewdie Birhanu, Megan C. Roberts, Victor Santana, Nickhill Bhakta.

**Project administration:** Diriba Fufa Hordofa, Yohannes Kebede.

**Software:** Yohannes Kebede.

**Supervision:** Yohannes Kebede, Zewdie Birhanu, Megan C. Roberts, Victor Santana, Nickhill Bhakta.

**Validation:** Diriba Fufa Hordofa, Yohannes Kebede, Zewdie Birhanu, Victor Santana, Nickhill Bhakta.

**Writing – original draft:** Diriba Fufa Hordofa, Yohannes Kebede, Mamude Dinkye, Tadele Hailu.

**Writing – review & editing:** Diriba Fufa Hordofa, Yohannes Kebede, Mamude Dinkye, Tadele Hailu, Zewdie Birhanu, Megan C. Roberts, Victor Santana, Nickhill Bhakta.

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
