## [Decision Letter · Decision Letter 0]

23 May 2025

PGPH-D-25-00293

Mapping Integrated Implementation of Adapted Resource and Implementation Application (ARIA) and REDCap version Hospital-Based Pediatric Cancer Registry (HBCR) in Ethiopia: An Implementation Study

Dear Dr. Hordofa,

Thank you for submitting your manuscript to PLOS Global Public Health. After careful consideration, we feel that it has merit but does not fully meet PLOS Global Public Health’s publication criteria as it currently stands. Therefore, we invite you to submit a revised version of the manuscript that addresses the points raised during the review process.

The manuscript has been evaluated by reviewers, and their comments are available below and in the attached files.

The reviewers have raised a number of concerns. They request improvements to the reporting of methodological aspects of the study, for example, regarding more information on how the data collection for the focus group was completed. One reviewer would like to see a more detailed description of thematic analysis findings including a figure, flowchart, or table as is traditional in this field. 

Could you please carefully revise the manuscript to address all comments raised?

We look forward to receiving your revised manuscript.

Kind regards,

Katherine Demi Kokkinias, Ph.D.

Staff Editor

Journal Requirements:

 2. In the ethics statement in the Methods, you have specified that verbal consent was obtained. Please provide additional details regarding how this consent was documented and witnessed, and state whether this was approved by the IRB

3. In the online submission form, you indicated that The data that support the findings of this study are available from the corresponding author, [DF], upon reasonable request. 

3. Uploaded as supplementary information.

4. We have noticed that you have uploaded Supporting Information files, but you have not included a list of legends. Please add a full list of legends for your Supporting Information files after the references list. 

Additional Editor Comments (if provided):

Reviewers' comments:

Reviewer's Responses to Questions

**Comments to the Author**

1. Does this manuscript meet PLOS Global Public Health’s publication criteria?

Reviewer #1: Yes

Reviewer #2: Yes

2. Has the statistical analysis been performed appropriately and rigorously?

Reviewer #1: I don't know

Reviewer #2: N/A

3. Have the authors made all data underlying the findings in their manuscript fully available (please refer to the Data Availability Statement at the start of the manuscript PDF file)?

Reviewer #1: No

Reviewer #2: Yes

4. Is the manuscript presented in an intelligible fashion and written in standard English?

Reviewer #1: Yes

Reviewer #2: Yes

Reviewer #1: Revisions requested:

The authors described how many interviews, discussions, and workshops were held but did not detail the number of personnel by role type in those engagements even though this was recorded on the interview guides. The data analysis section should also describe how key themes were discovered and validated.

Recommendation for future:

The authors documented future integrated workflows but could have described the current state and provided time estimates to use in a current-future state gap analysis to fully describe changes to current workflows and estimated time per case entry, especially if the new approach requires duplicate data entry into different IT systems (HIMS, REDCap, Aria).

Minor Issues:

• Lines 281-283 “ There are two interim documents created by the HBCR developing team to facilitate easy date collection and ensuring hard coy back. These are name pediatric oncology…” >> development team not developing team, data collection not date collection, and copy not coy, named not name

• Line 304: “the data clerk will collect the document and ensures the all the information is entered into the REDCap version HBCR after proper validation” >> ensures that all the information not ensures the all the

• Line 453: “Regular virtual discussion will be held among the core implementing team rom respective hospitals…” >> from not rom

• Line 593: “Redcap” should be REDCap

Reviewer #2: Journal: PLOS Global Public Health

ID: PGPH-D-25-00293

Title: Mapping Integrated Implementation of Adapted Resource and Implementation Application (ARIA) and REDCap version Hospital-Based Pediatric Cancer Registry (HBCR) in Ethiopia: An Implementation Study

Interesting manuscript, it is a descriptive and qualitative study; however, there are some observations that the authors should be considered:

Noteworthy that the authors did not use figures and tables to summarize results. Consequently, the results section is huge; it is recommended that the authors highlight or summarize the most important results in tables or even figures. For example, for each process or stages thereof, for each result of professional, chief and operative person who participated. It is also possible that in methodology for each process of stage of the implementation, authors can use flowcharts.

In general, on the discussion section it is necessary to compare and discuss their results with previous studies performed, but with other regional and international reports, it is not enough to just put the references, it is necessary to discuss with more detail.

References must be homogenized, for example in some cases the author's last name, without the name initials are used, and in other sections they do put both last name and initials. The limitations of the work are clearly missing.

And also recommended to summarize the conclusion.

It is recommended to check for editing errors such as double dots, spaces, etc

**Do you want your identity to be public for this peer review?** For information about this choice, including consent withdrawal, please see our Privacy Policy

Reviewer #1: No

Reviewer #2: **Yes: ** Leal Yelda A.

---

## [Decision Letter · Decision Letter 1]

20 Oct 2025

Mapping Integrated Implementation of Adapted Resource and Implementation Application (ARIA) and REDCap version Hospital-Based Pediatric Cancer Registry (HBCR) in Ethiopia: An Implementation Study

PGPH-D-25-00293R1

Dear Dr. Hordofa,

We are pleased to inform you that your manuscript 'Mapping Integrated Implementation of Adapted Resource and Implementation Application (ARIA) and REDCap version Hospital-Based Pediatric Cancer Registry (HBCR) in Ethiopia: An Implementation Study' has been provisionally accepted for publication in PLOS Global Public Health.

Best regards,

Julia Robinson

Executive Editor

Reviewer Comments (if any, and for reference):

Reviewer's Responses to Questions

**Comments to the Author**

Reviewer #2: All comments have been addressed

publication criteria?

Reviewer #2: Yes

3. Has the statistical analysis been performed appropriately and rigorously?

Reviewer #2: N/A

4. Have the authors made all data underlying the findings in their manuscript fully available (please refer to the Data Availability Statement at the start of the manuscript PDF file)?

Reviewer #2: Yes

5. Is the manuscript presented in an intelligible fashion and written in standard English?

Reviewer #2: Yes

Reviewer #2: The authors attended all of the recommendations and observations; I think the manuscript is OK for publication.

**Do you want your identity to be public for this peer review?** For information about this choice, including consent withdrawal, please see our Privacy Policy

Reviewer #2: No
